# Genome-Wide Identification of Cyclophilin Gene Family in Cotton and Expression Analysis of the Fibre Development in *Gossypium barbadense*

**DOI:** 10.3390/ijms20020349

**Published:** 2019-01-16

**Authors:** Qin Chen, Quan-Jia Chen, Guo-Qing Sun, Kai Zheng, Zheng-Pei Yao, Yu-Hui Han, Li-Ping Wang, Ya-Jie Duan, Dao-Qian Yu, Yan-Ying Qu

**Affiliations:** 1College of Agronomy, Xinjiang Agricultural University, Urumqi 830052, China; cqq0777@163.com (Q.C.); chqjia@126.com (Q.-J.C.); zhengkai555@126.com (K.Z.); yao_zp@163.com (Z.-P.Y.); hyh1461621613@163.com (Y.-H.H.); wlp010@126.com (L.-P.W.); dyj547525782@126.com (Y.-J.D.); yudaoqian88@163.com (D.-Q.Y.); 2Biotechnology Research Institute, Chinese Academy of Agricultural Sciences, Beijing 100081, China; sunguoqing02@caas.cn; 3Cotton Research Institute, Chinese Academy of Agricultural Sciences, Anyang 455000, China

**Keywords:** cyclophilins, genome-wide analysis, fibre, expression profiles

## Abstract

Cyclophilins (CYPs) are a member of the immunophilin superfamily (in addition to FKBPs and parvulins) and play a significant role in peptidyl-prolyl *cis*-*trans* isomerase (PPIase) activity. Previous studies have shown that CYPs have important functions in plants, but no genome-wide analysis of the cotton *CYP* gene family has been reported, and the specific biological function of this gene is still elusive. Based on the release of the cotton genome sequence, we identified 75, 78, 40 and 38 *CYP* gene sequences from *G. barbadense*, *G. hirsutum*, *G. arboreum*, and *G. raimondii*, respectively; 221 *CYP* genes were unequally located on chromosomes. Phylogenetic analysis showed that 231 *CYP* genes clustered into three major groups and eight subgroups. Collinearity analysis showed that segmental duplications played a significant role in the expansion of *CYP* members in cotton. There were light-responsiveness, abiotic-stress and hormone-response elements upstream of most of the CYPs. In addition, the motif composition analysis revealed that 49 cyclophilin proteins had extra domains, including TPR (tetratricopeptide repeat), coiled coil, U-box, RRM (RNA recognition motif), WD40 (RNA recognition motif) and zinc finger domains, along with the cyclophilin-like domain (CLD). The expression patterns based on qRT-PCR showed that six *CYP* expression levels showed greater differences between Xinhai21 (long fibres, *G. barbadense*) and Ashmon (short fibres, *G. barbadense*) at 10 and 20 days postanthesis (DPA). These results signified that *CYP* genes are involved in the elongation stage of cotton fibre development. This study provides a valuable resource for further investigations of *CYP* gene functions and molecular mechanisms in cotton.

## 1. Introduction

Cyclophilins (CYPs) are a member of the immunophilin superfamily (in addition to FKBPs and Parvulins) and are identified by their highly conserved cyclophilin-like domain (CLD) [1]. In 1984, Handschumacher et al. first identified cyclophilin A (CyPA) from bovine thymocytes, which is the receptor of the immunosuppressive drug cyclosporine A (CsA) [2]. Subsequently, an 18-kDa protein with PPIase (peptidyl-prolyl *cis*-*trans* isomerase) activity, which can catalyse the *cis*/*trans* isomerisation process of Xaa-proline peptide bonds, was found by Fischer and Takahashi et al. This process is a rate-limiting step in protein folding. Fischer and Takahashi et al. also discovered that PPIase is the cyclosporin A-binding protein cyclophilin [3,4]. *CYPs* are ubiquitous proteins present in bacteria, plants and animals [5,6]. To date, genome-wide analysis has been used to identify 19 *CYPs* in the human genome, along with 6 *CYPs* in *Escherichia coli*, 8 *CYPs* in *Saccharomyces cerevisiae*, 9 *CYPs* in *Schizosaccharomyces*, 14 *CYPs* in *Drosophila melanogaster*, 17 *CYPs* in *Caenorhabditis elegans* and 12 *CYPs* in *Leptosphaeria maculans* [7,8]. In contrast, plants have a higher number of *CYPs*, with 29 in *Arabidopsis thaliana*, 27 in rice, 62 in soybean, 77 in *B. napus* and 39 in maize [9,10,11,12,13].

There are more studies on the classification, physiological function and molecular mechanism of human *CYPs* than of plant *CYPs*, for which few studies are available [14]. The first plant cyclophilins were described successively in tomato (*Lycopersicon esculentum*), maize (*Zea mays*), and oilseed rape (*Brassica napus*) in 1990 [15]. Recently, studies have shown that OsCYP2 interacts with a C2HC-type zinc finger protein (OsZFP) participating in auxin signalling pathways controlling lateral root development [16], and soybean cyclophilin GmCYP1 interacts with an isoflavonoid regulator GmMYB176, which is induced by various abiotic stresses [17]. The reports also revealed that the AtCYP20-3 mutant is hypersensitive to oxidative stress conditions [18]. Two *Brassica rapa* cyclophilin genes, *ROC1-1* and *ROC1-2*, were involved in the light induction response [19]. Moreover, *CYP1* acts as a virulence determinant in rice blasts, and *CYP1* mutants show reduced virulence and are impaired in associated functions [20]. Furthermore, along with being involved in receptor signalling, abiotic stresses, photosynthetic reactions and immunisation against pathogens, *CYPs* play major roles in apoptosis, RNA processing and spliceosome assembly [18,21,22].

Cotton fibres are highly elongated from single-celled seed coats; they are not only important raw materials for the textile industry but also serve as ideal experimental materials for the study of cell elongation, cell wall formation and cellulose synthesis [23]. Two cultivated species of tetraploid cotton, *G. hirsutum* and *G. barbadense*, originated from a natural hybrid between A-genome cotton (such as *G. arboreum*) and D-genome cotton (such as *G. raimondii*) that occurred approximately 1–1.5 million years ago [24,25]. One of the major breeding targets of these cotton species is to improve fibre quality.

Recently, as more and more plant genome data have been published, genome-wide identification and analysis has become an effective method for rapidly predicting gene functions from a large family of genes. There are an increasing number of reports about plant gene families. For example, genome-wide identification and expression analysis of the chitinase gene family in Brassica rapa revealed that it plays a crucial role in the pathogen resistance of the host plants [26]; genome-wide identification and analysis of japonica and indica rice cultivars identified 502 candidate low-temperature-responsive genes and proposed a model that includes a pathway for cold stress-responsive signalling [27]; genome-wide analysis of gene expression profiling indicated that the COP9 signalosome is essential for correct expression of Fe homeostasis genes in Arabidopsis [28]; genome-wide analysis of small RNAs revealed 257 novel microRNAs related to cotton fibre elongation [29]; and genome-wide analysis of the WD40 protein family revealed that WD40 is involved in cotton fibre development [30]. However, little is known about the identification and function of the cyclophilin gene family in *Gossypium*. In particular, the association of this gene family with fibre development is unknown. Based on the cotton genome sequence database, we first identified 75, 78, 40 and 38 *CYP* genes from *G. barbadense*, *G. hirsutum*, *G. arboreum* and *G. raimondii*, respectively, and then investigated the expression patterns of *GbCYPs* in cotton fibre. These results provide a valuable resource for further investigation of the functions and molecular mechanisms of the *CYP* family in cotton.

## 2. Results

### 2.1. Identification of CYP Genes in Four Gossypium Species Based on the Conserved CLD Domain

Using the Hidden Markov Model (HMM) profile of the PPIase domain (PF00160) and *CYP* sequences from *Arabidopsis* (29), rice (27) and *B. napus* (77) as queries, a total of 80, 83, 42 and 38 predicted *CYP* gene sequences were identified from *G. barbadense*, *G. hirsutum*, *G. arboreum* and *G. raimondii*, respectively. The putative CYP members were then analysed by SMART and CDD databases to determine whether there was a complete or partially conserved CLD motif, removing sequences below the significance threshold. Finally, 75, 78, 40 and 38 *CYP* genes were identified from *G. barbadense*, *G. hirsutum*, *G. arboreum*, and *G. raimondii*, respectively. According to the naming method of *AtCYPs* by He and Romano et al. [9,31], 231 *CYP* genes encoding proteins from cotton were named by their relative molecular weight; we used immature proteins because different tools predict different cleavable signal peptides. Furthermore, proteins with similar molecular weights were consecutively numbered. The predicted *CYP* genes encoded proteins ranging between 64 and 185 amino acids, with predicted molecular weights ranging between 8.50 and 142.50 kDa. The minimum isoelectric point was 4.5 (GbCYP40-2), and the maximum isoelectric point was 11.5. Statistics showed that 62.3% of proteins were basic proteins. Of these, 120 cyclophilin proteins were localised in the cytoplasm, and the rest were localised in the nucleus, chloroplast, mitochondria, plasma membrane and extracellular tissues. Detailed information about the *CYP* gene family in cotton is displayed in Appendix A: Appendix A.

### 2.2. Phylogenetic Analysis

To analyse the evolutionary relationship of *CYP* family genes, we constructed an unrooted neighbour-joining (NJ) phylogenetic tree by the multiple alignment of 260 CYP proteins, comprising 29 AtCYPs from *A. thaliana* and 231 cotton CYPs from *G. barbadense*, *G. hirsutum*, *G. arboretum* and *G. raimondii* (Figure 1). The results showed that 182 cotton CYP proteins appeared in pairs (91 pairs), and every six were clustered together (2 GbCYPs, 2 GhCYPs, 1 GaCYP and 1 GrCYP). These proteins were classified into three major groups according to their genetic relationship, and Group I contained six subgroups, Ia, Ib, Ic, Id, Ie and If, similar to the *Arabidopsis* evolutionary tree grouping reported by Romano et al. [9,31]. Nine genes, *GbCYP19-1*, *GbCYP20-3*, *GbCYP70*, *GhCYP20-4*, *GhCYP30-1*, *GhCYP89-1*, *GaCYP18-2*, *GaCYP20-2* and *GaCYP89-1*, only appeared in the cotton A-subgenome, while *GbCYP18-2*, *GrCYP18-2* and *GhCYP18-2* only appeared in the D-subgenome. *GbCYP14-1*, *GbCYP14-2*, *GbCYP21-1*, *GbCYP22*, *GbCYP47-1* and *GbCYP63* were endemic to island cotton. *GhCYP12* and *GhCYP19-1* only appeared in the upland cotton genome. Additionally, *GaCYP15* and *GrCYP23-1* were only in *G. arboretum* and *G. raimondii*, respectively. *GhCYP18-1*, *GaCYP18-1* and *GrCYP18-1* are homologous genes, but they were lost or mutated in island cotton. These 25 genes were far from other members of the *CYP* family in cotton.

### 2.3. Gene Structure and Domain Analysis

To obtain the gene structures of the cotton *CYP* genes, the intron/exon structures of 231 *CYP* genes were analysed by the GSDS online server, as displayed in Figure 2C and Appendix A: Appendix A. The analysis showed that 34 genes (*GbCYP14-2*, *GbCYP16-1*, *GbCYP18-1* to *GbCYP18-9*, *GhCYP12*, *GhCYP18-2* to *GhCYP18-9*, *GhCYP18-11*, *GhCYP18-12*, *GaCYP15*, *GaCYP18-3* to *GaCYP18-7* and *GrCYP18-2* to *GrCYP18-7*) had only one exon, and there were 21 exons in *GbCYP142*, which had the largest number of exons. Conserved motif prediction showed that cyclophilins could be divided into single-domain (SD) and multi-domain (MD) groups. Among the 231 *CYP* genes, 182 genes encoded a single CLD domain, and the remaining 49 *CYPs* possessed other functional domains, such as TPR, coiled coil, U-box, RRM, WD40 and zinc finger domains (Figure 3). The CLD domain was composed of motifs 1, 2, 3, 4, 5, 6 and 7 (Figure 2B, Appendix A: Appendix A, Appendix A). In contrast to the results of the phylogenetic analysis, most members belonging to the same subgroup had similar gene structures and functional domains.

### 2.4. Chromosomal Locations and Gene Collinearity Analysis

To determine the distributions of *CYP* genes on different chromosomes of cotton, we constructed a chromosome map with the information from four cotton genomes (Figure 4, Appendix A: Appendix A). The results showed that most of these genes (221) were unequally located on 13 chromosomes, while 8 genes, *GrCYP27*, *GrCYP42-1*, *GrCYP57*, *GrCYP72-2*, *GrCYP90*, *GhCYP12*, *GhCYP18-10*, *GhCYP28-1*, *GhCYP49-2* and *GhCYP49-3*, were located on different scaffolds. The number of genes distributed on each chromosome was different; A2-chr7 and D5-chr8 had the largest number of genes (6), while only one gene was found on A2-chr8, A2-chr10, D5-chr3, D5-chr4, AD1-D05 and AD2-D06. No CYP gene was found on the AD2-D05 chromosome of sea-island cotton.

To reveal the expansion mechanism of the *CYP* gene family, all intragenomic and intergenomic duplication data files of *G. barbadense* and three other *Gossypium* species were filtered by MCScanX. In total, we identified 8 pairs of tandem duplications (GbCYP63/GbCYP77, GbCYP66-1/GbCYP70, GhCYP18-8/GhCYP18-7, GhCYP89-1/GhCYP92, GhCYP89-2/GhCYP91, GaCYP18-6/GaCYP18-5, GaCYP89-1/GaCYP89-2 and GrCYP18-5/GrCYP18- 6) in the four cotton species, 39 pairs of segmental duplications in *G. barbadense* (Figure 4, Appendix A: Appendix A), and 117, 51 and 83 intergenomic duplications between *G. barbadense* and *G. hirsutum*, *G. arboretum* and *G. raimondii*, respectively (Figure 5). The details for the collinear gene pairs are listed in Appendix A: Appendix A. Among the 75 *GbCYP* genes, we identified 37 pairs of common orthologous *CYP genes* in the four Gossypium species. Using the nonsynonymous (Ka) and synonymous (Ks) substitution rates of each duplicated *CYP* gene pair, we ascertained that the evolutionary selection pressures of *CYP* genes in the four *Gossypium* species tended to be purified, and the structures of these genes, except for GbCYP18-1, GbCYP29, GbCYP40-3, GbCYP43-1, GbCYP47-2, GbCYP58, GhCYP18-4, GhCYP40-2, GhCYP45-2, GhCYP70-1, GaCYP31, GaCYP70 and GrCYP40-3, were highly conserved during evolution (Appendix A).

### 2.5. Cis-Element Analysis of CYP Promoter Sequences

To analyse the *CYP* gene *cis*-acting element, we retrieved the 2000 bp upstream of the 75 *GbCYP* genes from the initiation codon. The statistics showed that all of the *GbCYP* genes included at least one of the 16 putative light-responsiveness elements (3-AF1, ACE, AE-box, chs-CMA1a, GA-motif, GATA-motif, G-box, GT1-motif, I-box, LAMP, Sp1, TCCC-motif, TCT-motif, ATCT-motif, Box 4 and MRE). In particular, 65 *GbCYP* genes included Box 4 elements (part of a conserved DNA module involved in light responsiveness), and 55 members contained the light-responsiveness element GT1-motif. Furthermore, the putative promoter region of 34 members included low-temperature-responsiveness elements (LTRs), 28 *GbCYPs* included drought- inducible elements (MBSs), and 21 *GbCYPs* had *cis*-acting elements involved in defence and stress responsiveness (TC-rich repeats). Moreover, eight types of hormone-related-response elements were identified: abscisic-acid-responsiveness (ABRE), auxin-responsiveness (AuxRR-core), gibberellin-responsiveness (GARE-motif, P-box), salicylic-acid-responsiveness (TCA-element) and MeJA-responsiveness (CGTCA-motif) elements (Figure 6, Appendix A: Appendix A).

### 2.6. Gene Expression Analysis

Based on the RNA-seq transcriptome data, the RPKMs of different tissues (seeds, stems, leaves and flowers) and fibre developmental stages (0 DPA ovules and 5 DPA, 10 DPA and 15 DPA fibres) were used to investigate the expression profiles of *CYP* family genes in sea-island cotton (Appendix A: Appendix A). The heat map shows that these *GbCYP* genes clustered in six pattern groups. In general, genes within the same subgroup showed similar expression patterns (Figure 7). The first group included *GbCYP8*, *GbCYP21-1* and *GbCYP37-2*, which were highly expressed in seeds. In the second group, 14 genes were highly expressed in 5 and 10 DPA fibres. In the third group, 14 genes were highly expressed in leaves, while 4 genes were highly expressed in stems or flowers in the fourth group. The other genes in groups five and six were highly expressed in 0 DPA ovules. The various expression patterns suggested the functional divergence of different groups of *GbCYP* members in cotton.

To further study the relationship between *GbCYPs* and cotton fibre development, we isolated 35 genes with RPKM ≥ 5 at 5, 10 and 20 DPA for further real-time quantitative RT-PCR (qRT-PCR) analysis. Of these, 15 genes were highly expressed in 5 DPA fibres (Figure 8A and Figure 9), and *GbCYP18-6*, *GbCYP27-2*, *GbCYP28*, *GbCYP37-4* and *GbCYP37-5* were highly expressed in 10 DPA fibres (Figure 8B and Figure 9). *GbCYP24-1* and *GbCYP26-3* were highly expressed in 15 DPA fibres (Figure 8B). In addition, 13 genes were mainly expressed in 20 DPA fibres (Figure 8C and Figure 9). The results of qRT-PCR proved that the fold-change values of *GbCYPs* were reliable. It is notable that differentially expressed *GbCYPs* were mainly distributed in the second group.

Additionally, Xinjiang is the only long-staple cotton-producing area in China. Xinhai21 is a hybrid of the Giza variety from Egypt and has been the major cultivar in Xinjiang over the years, and Ashmon is a resource material that was introduced from Syria. Our research team investigated the quality of these two sea-island cotton fibres over the three years under field conditions and found that Xinhai21 had a greater fibre length than Ashmon (Table 1) [32]. To understand the role of *GbCYPs* during different developmental stages of cotton fibre, we chose Ashmon as the control material and analysed the expression profiles of six *GbCYPs* (*GbCYP18-5*, *GbCYP18-6*, *GbCYP27-2*, *GbCYP36*, *GbCYP47-1* and *GbCYP58*) between Xinhai21 and Ashmon. As shown in Figure 9, there were significant differences in some fibre development stages. Generally, the expression levels of six Xinhai21 *CYPs* were higher than those of the same six Ashmon *CYPs*. *GbCYP47-1* showed a progressive decrease from 5 DPA to 25 DPA in two cotton species, and *GbCYP18-6* showed the highest overall expression levels at 10 DPA and 15 DPA and a 5-fold increase between 5 DPA and 10 DPA. The expression of *GbCYP27-2* peaked at 10 DPA, and the expression levels of *GbCYP18-5*, *GbCYP36* and *GbCYP58* peaked at 20 DPA.

## 3. Discussion

### 3.1. Identification of the CYP Gene Family

In this study, we identified a total of 75, 78, 40 and 38 *CYP* genes from *G. barbadense*, *G. hirsutum*, *G. arboreum*, and *G. raimondii*, respectively. The number of *CYP* genes in tetraploid cotton was approximately twice that in diploid cotton. The evolutionary tree showed that more CYP proteins appeared in pairs and clustered together with 2 GbCYPs, 2 GhCYPs, 1 GaCYP and 1 GrCYP, which supported the cotton species polyploidisation event that occurred 1.5 million years ago [24,25]. Moreover, there were 25 genes far from other members of the *CYP* family in cotton, implying that these *CYP* family genes underwent different evolutionary processes. Some newly generated genes and pre-existing genes might have been deleted during cotton evolution. The chromosomal locations and collinearity showed that segmental duplications played a significant role in the expansion of *CYP* members in cotton, which was the same as the results reported in *A. thaliana*, rice, *B. napus* and other plants [9,10,11,12,13]. Approximately half (49.3%) of the common orthologous *CYP genes* are shared by all four Gossypium species, indicating that these collinear pairs might have already existed before the genome rearrangement. The promoter analysis results suggest that *GbCYPs* may play an important role in responses to light, stress and hormone signalling.

### 3.2. Gene Function Analysis Revealed CYP Contributions to Various Cellular Processes

Cyclophilins are present in all subcellular compartments, are involved in various metabolic processes, especially protein interactions, and act as molecular chaperones [5,33]. The CLD has an overall beta-barrel structure, which consists of eight antiparallel beta-strands and two alpha-helices and catalyses a rate-limiting step of the *cis*/*trans* isomerisation process in protein folding [34]. Interestingly, cyclophilins contain some special domains in addition to the CLD, suggesting that cyclophilin proteins have various biological functions in cotton.

As shown in Figure 3, we identified 28 cotton CYPs containing an additional tetratricopeptide repeat (TRP) domain at the C-terminus, and these CYPs were highly homologous to AtCYP40 (Appendix A: Appendix A). The TPR motif consists of a repetitive 34-amino acid sequence that is essential for AtCYP40 binding to Hsp90 (the heat shock protein 90). The TPR domain can regulate Hsp90 ATPase activity and aid in protein folding, trafficking and the assembly of multiprotein complexes [35]. Moreover, a complex of AGO1, HSP90, CYP40, with a small RNA duplex is a key intermediate of RNA-induced silencing complex (RISC) assembly and participates in posttranscriptional gene silencing [36]. The mutants of AtCYP40 show development defects [37]. Thus, we speculated that the 28 cotton CYPs and AtCYP40 may have similar functions.

In contrast, GaCYP70, GrCYP63, GhCYP70-1, GhCYP70-2 and GbCYP58 contained four WD40 (tryptophan-aspartate) repeats at the N-terminus and were highly homologous to AtCYP71. WD40 domains, which were originally identified in G-proteins, widely participate in protein–protein interactions, signal transduction and transcription regulation [38] and are involved in cotton fibre development during the initiation and elongation stages [30,39]. In addition, AtCYP71 interacts with histone H3 and is involved in chromatin assembly, determining the development of *Arabidopsis* [40].

Six cotton CYPs (GaCYP47, GrCYP47, GhCYP47, GhCYP48, GbCYP40-2 and GbCYP61) possess a coiled coil (CC) domain; this domain derives its name from a specific α-helix–α-helix interaction and is a modulator of downstream responses involved in immune signalling [41,42]. There are also four CYPs (GaCYP65, GrCYP65, GhCYP65-1 and GhCYP65-2) that contain a U-box motif and play a general role in ubiquitination [43]. Furthermore, GrCYP72-1, GhCYP70-3, GhCYP70-4, GbCYP47-1 and GbCYP79 contain an RNA recognition motif (RRM) and a CCHC-type (Zinc finger) motif and are homologous to AtCYP59. We called these types of cyclophilins cyclophilin-RNA interacting proteins (CRIPs). Loss of these proteins has been shown to affect transcription regulation and the subunit phosphorylation of RNA polymerase II [44].

Lastly, GbCYP142, the largest CYP in *Gossypium*, contains a transmembrane domain (TR) and two ribonuclease MRP protein subunits (POP1/POPLD), suggesting that GbCYP142 might be involved in the metabolism of RNA molecules [45]. In addition, GbCYP142, AtCYP21-4 and OsCYP21-4 are homologous genes. Research has shown that OsCYP21-4 expression levels are increased in response to various abiotic stresses [46]. These results underline the important functions of *CYPs* in many essential physiological processes of cotton growth, including fibre development.

### 3.3. Expression Analysis Revealed CYPs Involved in Cotton Fibre Development

Of the two tetraploid cultivated *Gossypium* species, *G. barbadense* is prized for its extra-long fibres. The RPKM values of various expression patterns from different *G. barbadense* tissues showed that lost *CYPs* were highly expressed in ovules and fibres. qRT-PCR showed that 35 *GbCYPs* were highly expressed in 5–20 DPA fibres. Furthermore, the expression levels of six Xinhai21 *CYPs* were higher than those of Ashmon *CYPs*. These results signify the potential function of CYPs in the elongation stage of cotton fibre development.

Recently, research has confirmed that the phloem constitutes a long-distance transport system for trafficking plant hormones and signals and for the distribution of nutrients throughout the plant. Giavalisco et al. identified 140 soluble phloem proteins of *B. napus* by 1-DE and high-resolution 2-DE, which included 9 CYP proteins [47], and Hanhart et al. performed LC-MS/MS analysis of phloem protein extract of *B. napus* and found 12 BnCYPs in the phloem [11]. Investigating the *dgt* (auxin-resistant diageotropica) mutant of tomato and using Fourier transform infrared spectroscopy revealed that cyclophilin (*LeCYP*) is a component of auxin signalling [48]. Yeast two-hybrid and pull-down assays demonstrated that *AtCYP20-2* interacted with BZR1 and changed its conformation to regulate the activity of Brassinolide (BR) [49]. BR and auxin are integrant elements for cotton fibre initiation and elongation [50,51]. These results indicate that some cyclophilins probably have essential functions in long-distance hormone signalling that are involved in cotton fibre development.

In addition, Ramsay et al. reported that WD40 repeat proteins interacted with MYB and bHLH in a double direction to regulate cotton fibre development during the initiation and elongation stages [39]. We found that 5 CYPs contained WD40 repeats (Figure 3), and real-time qRT-PCR also confirmed that *GbCYP58* was more highly expressed in Xinhai21 than in Ashmon. This finding indicates that some cyclophilins may be involved in cotton fibre development through protein-protein interactions. However, the real molecular mechanisms between cyclophilins and cotton fibre remain to be further studied.

## 4. Methods

### 4.1. Identification and Sequence Analysis of CYPs

We downloaded the *G. arboreum* (BGI, version 2.0), *G. raimondii*, (BGI, version1.0), and *G. barbadense* (HAU_v2.0) data from CottonGen (https://www. cottongen.org/). *G. hirsutum* acc. TM-1 and *G. barbadense* acc. XinHai21 were downloaded from http://mascotton.njau.edu.cn and http://www.chgc.sh.cn/, respectively, and the *A. thaliana* genome sequence data were from Phytozome v11.0 (http://www.phytozome.net/). The HMM profile of the cyclophilin-like domain (CLD, PF00160) was downloaded from the Pfam website (http://pfam.xfam.org/) and was used as the query to identify all possible cyclophilin-like sequences with HMMER software (http://hmmer.org). Furthermore, CDD (http://www.ncbi.nlm.nih.gov/cdd), Pfam (http://pfam.xfam.org/) and SMART (http://smart.embl-heidelberg.de/) databases were used to confirm the conserved CLD domain of the candidate sequence. The biophysical properties of the CYP proteins were calculated using the ExPASy online server tool (https://www.expasy.org/). These *CYP* genes were named on the basis of their molecular weight.

### 4.2. Phylogenetic and Possible Functional Analysis

Multiple protein sequence alignment of the CLD domains from *G. raimondii*, *G. arboretum*, *G. hirsutum*, *G. barbadense* and *Arabidopsis* was performed by ClustalW2 (http://www.genome.jp/tools- bin/clustalw), and the alignment was imported into MEGA 5.0 (Center for Evolutionary Medicine and Informatics, Arizona State University, Tempe, AZ, USA) with pairwise distance and the NJ algorithm to construct a phylogenetic tree using the p-distance method with 1000 bootstrap replications [52]. The exon/intron structure of the cotton *CYP* genes was analysed by GSDS 2.0 (http://gsds.cbi.pku.edu.cn/). The MEME programme (http://meme-suite.org/) was employed to identify the conserved motifs in the *CYP* sequences, with the following parameters: number of unique motifs: 15; maximum and minimum search widths: 32; and 6 amino acid residues. The promoter sequences (2 kb upstream from the translation start site) of the *CYP* genes were obtained from the *G. barbadense* genomes, and 75 *GbCYP* gene *cis*-acting regulatory elements were predicted by PlantCARE (http://bioinformatics.psb.ugent.be/webtools/plantcare/html/).

### 4.3. Chromosomal Locations and Gene Collinearity Analysis

The physical chromosome locations of all *CYPs* were obtained from the gff3 files of four cotton species databases. Mapchart 2.0 (https://mapchart.net/) software was adopted to visually map the chromosomal location [53]. Gene duplication events were analysed using the Multiple Collinearity Scan toolkit (MCScanX: http://chibba.pgml.uga.edu/mcscan2/) [54]. To exhibit segmentally duplicated pairs and orthologous pairs of *CYP* genes from *G. barbadense* and the three other *Gossypium* species, we used Dual Systeny Plotter software (https://github.com/CJ-Chen/TBtools) to draw collinearity maps. Thereafter, the synonymous (Ks) and nonsynonymous (Ka) substitution rates of each duplicated *CYP* gene were estimated by the PAL2NAL programme (http://www.bork.embl.de/pal2nal/).

### 4.4. Gene Expression Analysis

The reads per kb per million reads (RPKM) values were obtained from the transcriptome data of *Gossypium barbadense* cv. Xinhai21 [25], which were provided by Tian-Zhen Zhang and his research group. We isolated the *GbCYPs* with RPKM ≥ 5 for further expression analysis.

### 4.5. RNA Isolation and qRT-PCR Analysis

Total RNA was isolated using the RNAprep Pure Plant Kit (TIARGEN, Beijing, China) and was treated with DNase I to remove genomic DNA. The RNA quality and purity were measured by a NanoDrop 2000 spectrophotometer (Thermo Scientific, massachusetts, USA), and 1 μg of total RNA was used to synthesise first-strand cDNA with the Transcriptor First Strand cDNA Synthesis Kit and oligo-dT primers at 42 °C for 60 min and 72 °C for 10 min. Real-time PCR was performed in a 7500 Fast Real-Time PCR system (Applied Biosystems, State of California, USA) using SYBR Green Master Mix. Three biological replicates were performed per cDNA sample, and each reaction was prepared in a total volume of 20 μl containing 10 μL of SYBR Green PCR Master Mix, 1 μL of each primer, 2 µL of diluted cDNA template, and 6 µL of ddH_2_O. The cotton *UBQ7* gene was used as a reference, and the PCR conditions were as follows: 95 °C for 5 min followed by 40 cycles of 95 °C for 5 s and 60 °C for 34 s. The 36 primers (Appendix A) were designed by Primer Premier 5, and the results were analysed with the 2^−ΔΔ*C*t^ method.

## 5. Conclusions

CYPs are ubiquitous proteins that are found in bacteria, plants and animals. There are a large number of *CYPs* in plants, but the functions of most plant *CYPs* are still elusive. For the first time, we performed a genome-wide analysis of *CYP* gene families in four *Gossypium* species. A total of 231 *CYP* genes were retrieved based on the conserved CLD domain and were clustered into three major groups and eight subgroups. Subsequently, a detailed analysis of their phylogeny, structures, conserved motifs, chromosomal locations, gene collinearity, *cis*-acting elements, and expression patterns in different fibre development stages was performed. The results revealed that most cyclophilin proteins are SD proteins, segmental duplications that played a significant role in the expansion of CYP members, and 52% of all CYPs are localised in the cytoplasm in cotton. Cyclophilin proteins may be involved in various processes of cotton growth and development. The expression patterns observed via qRT-PCR analysis showed that the expression levels of six Xinhai21 (long fibre) *CYPs* were higher than those of Ashmon (short fibre) *CYPs*, and *GbCYP* genes may be involved in the elongation stage of cotton fibre development. In addition, these results provide a valuable resource for further investigation of *CYP* gene functions in cotton.

## Figures and Tables

**Figure 1 ijms-20-00349-f001:**
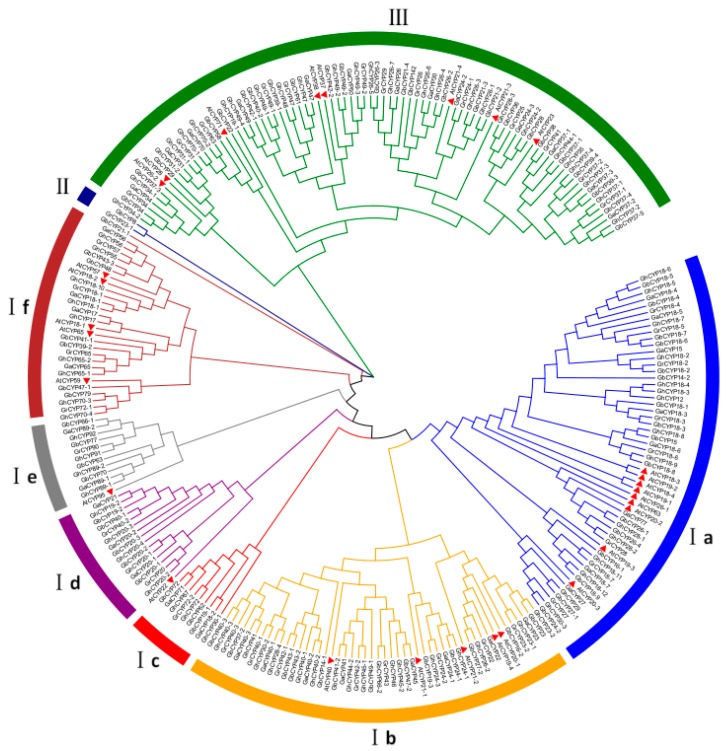
Phylogenetic tree of 260 cyclophilin (CYP) proteins from *G. barbadense* (A-subgenome 42, D-subgenome 33), *G. hirsutum* (A-subgenome 41, D-subgenome 37), *G. arboretum* (40), *G. raimondii* (38) and *Arabidopsis* (29). MEGA 6.0 was used to build the neighbour-joining (NJ) tree with 1000 bootstrap replicates. Different line colours indicate different subgroups of CYPs. AtCYPs are represented by red triangles.

**Figure 2 ijms-20-00349-f002:**
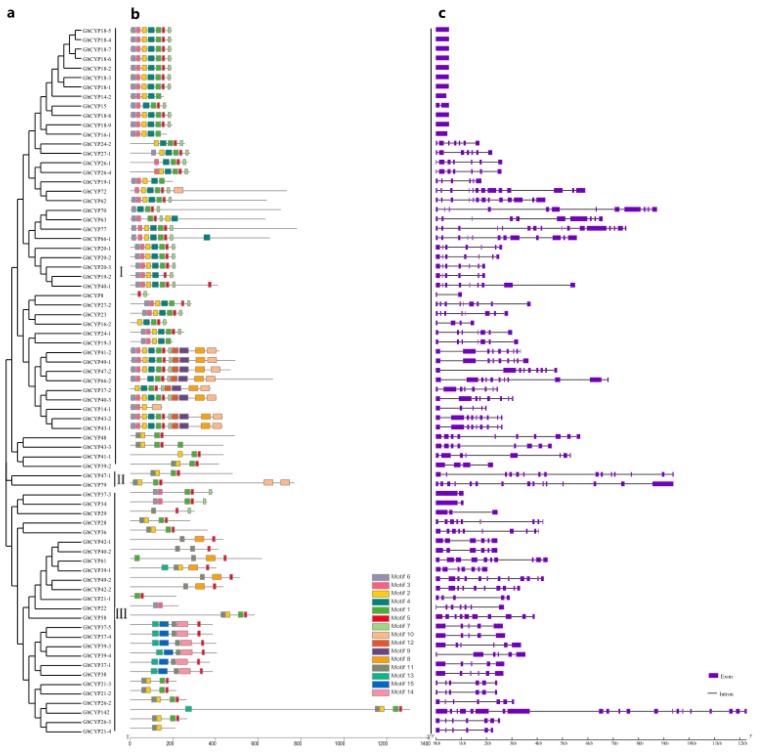
(**a**) Phylogenetic relationship, (**b**) motif analysis and (**c**) exon-intron gene structures of the *CYP* gene family in *G. barbadense*. a phylogenetic analysis of GbCYPs using the NJ method. b Motifs 1, 2, 3, 4, 5, 6 and 7 together compose the cyclophilin-like domain (CLD). Motif 8 is a TPR (tetratricopeptide repeat) domain function. c Structures of *CYP* genes.

**Figure 3 ijms-20-00349-f003:**
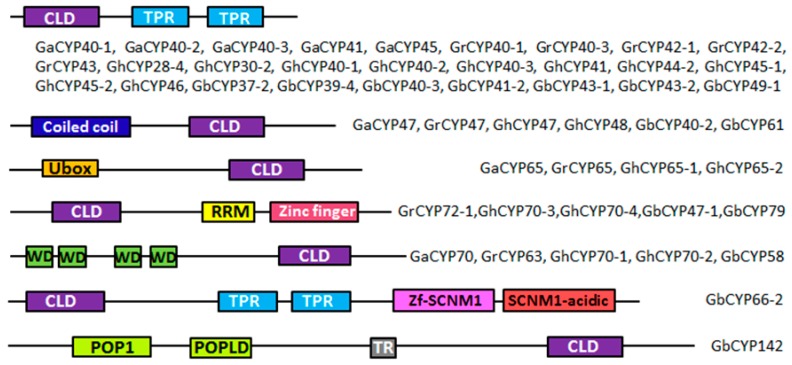
MD structures of 49 *Gossypium* CYPs. CLD = cyclophilin-like domain; TPR = tetratricopeptide repeat; U-box = U-box domain; RRM = RNA recognition motif; WD = tryptophan-aspartate repeat; Zf = Zinc finger of sodium channel modifier 1; POP1/POPLD = MRP protein subunit; TR = transmembrane domain.

**Figure 4 ijms-20-00349-f004:**
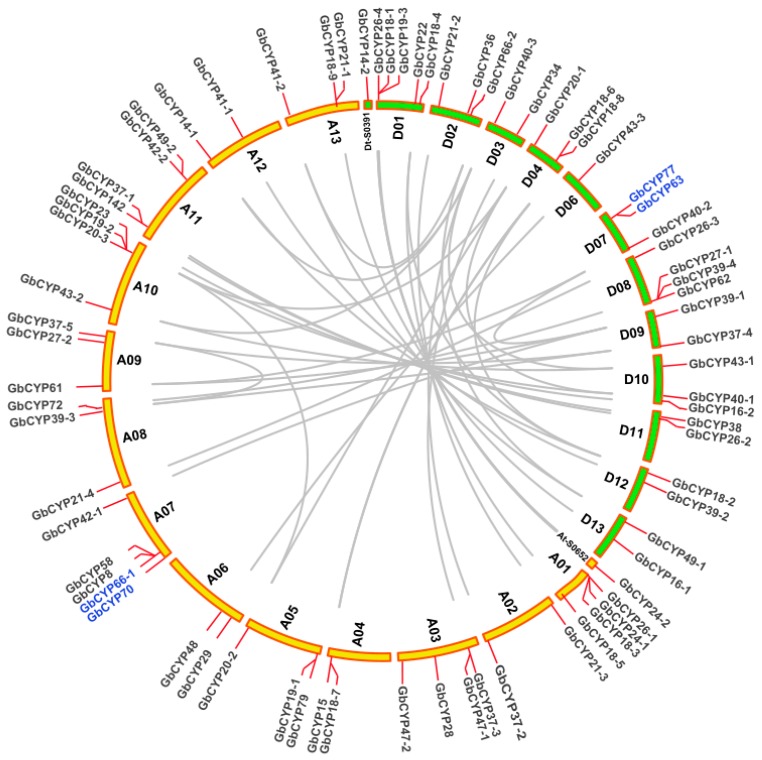
Chromosomal localisations and duplications of *CYP* genes on *G. barbadense* chromosomes. The duplicated *CYP* genes are indicated with grey lines and tandemly duplicated genes are marked with blue.

**Figure 5 ijms-20-00349-f005:**
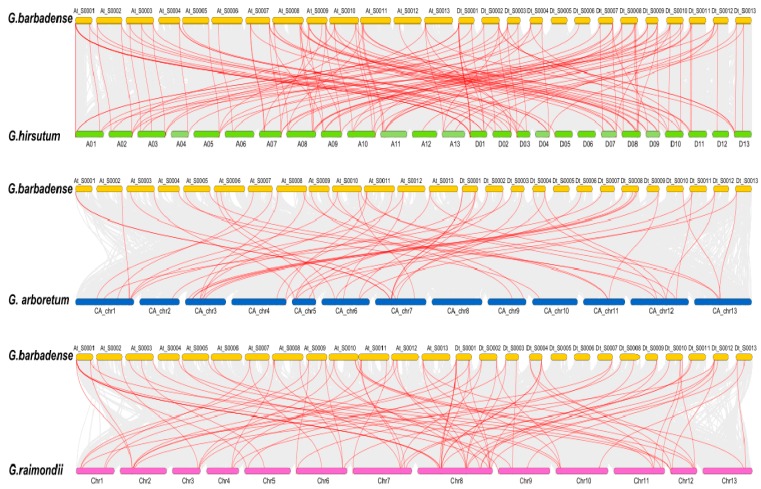
Collinearity analyses of *CYP* genes between *G. barbadense* and *G. hirsutum*, *G. arboretum* and *G. raimondii*. Duplication events within *G. barbadense* and the genomes of the other three *Gossypium* species are indicated with grey lines in the background, while the red lines highlight syntenic *CYP* gene pairs.

**Figure 6 ijms-20-00349-f006:**
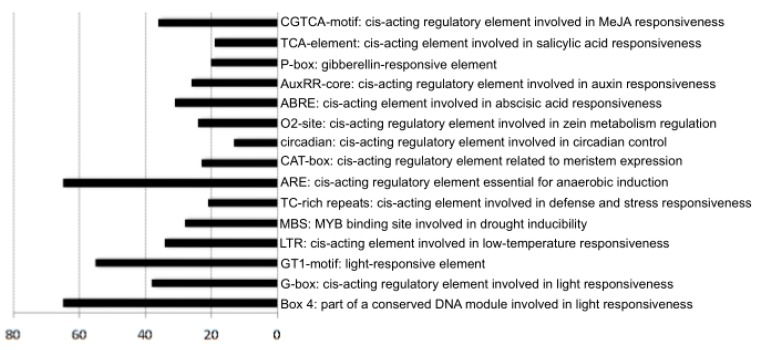
The main response *cis*-acting elements in *GbCYP* promoters.

**Figure 7 ijms-20-00349-f007:**
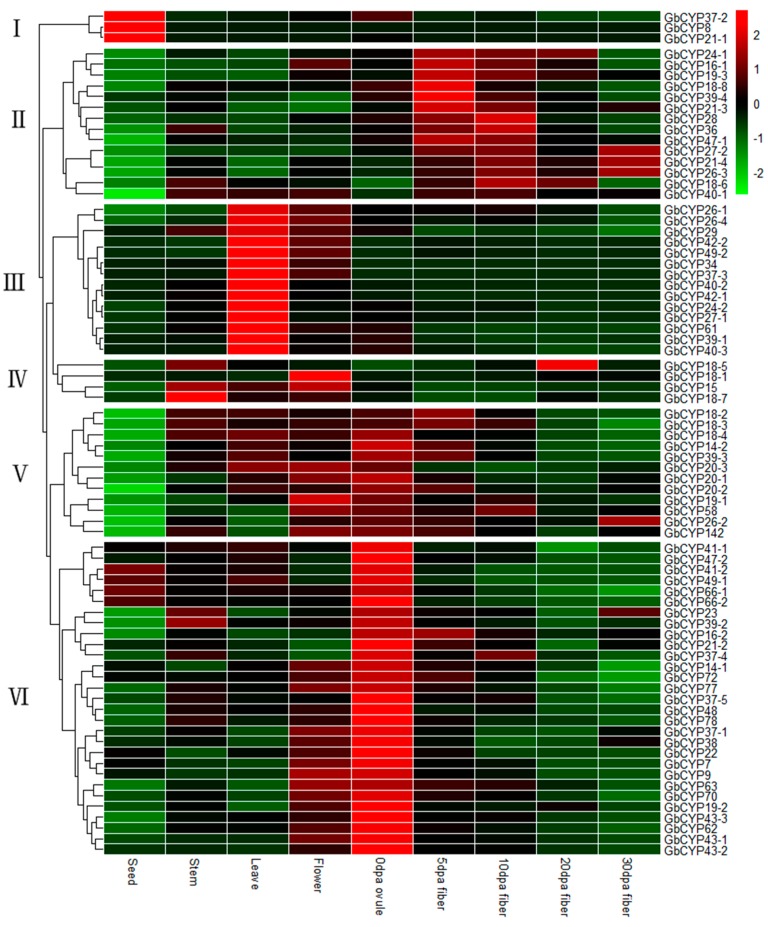
Expression patterns of *GbCYP* genes in different tissues. Nine tissues comprising seeds, stems, leaves, flowers, 0 days postanthesis (DPA) ovules, and 5, 10, 20 and 30 DPA fibres were investigated. Scale bars represent the log2 transformations of the RPKM values. The results can be classified into six pattern groups (I–VI).

**Figure 8 ijms-20-00349-f008:**
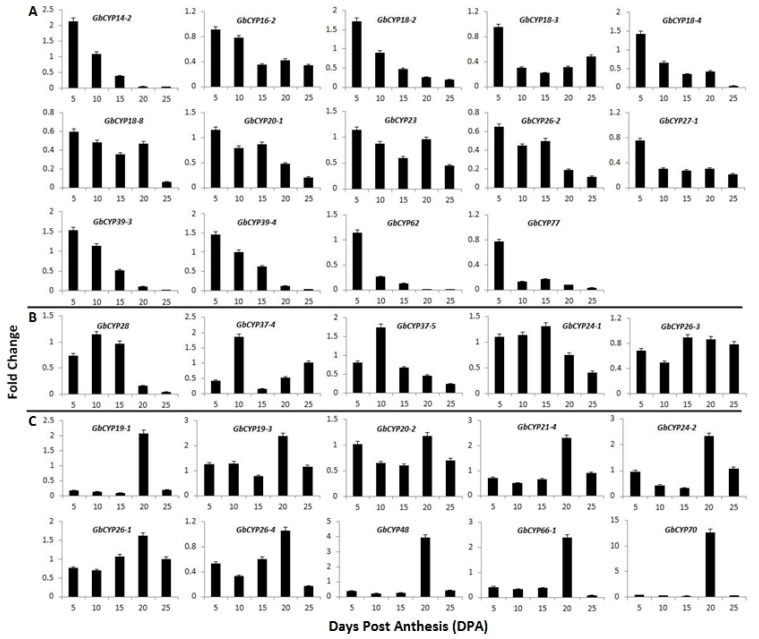
Expression patterns of 29 *G. barbadense CYP* genes during fibre development based on qRT-PCR. (**A**) Genes that were highly expressed at 5 DPA. (**B**) Genes that were highly expressed at 10 and 15 DPA. (**C**) Genes that were highly expressed at 20 DPA. *GbUBQ7* was used as an internal reference to normalise the expression data. Error bars were calculated from the difference in the expression patterns of three biological replicates. The details of the primer sequences are listed in Appendix A: Appendix A.

**Figure 9 ijms-20-00349-f009:**
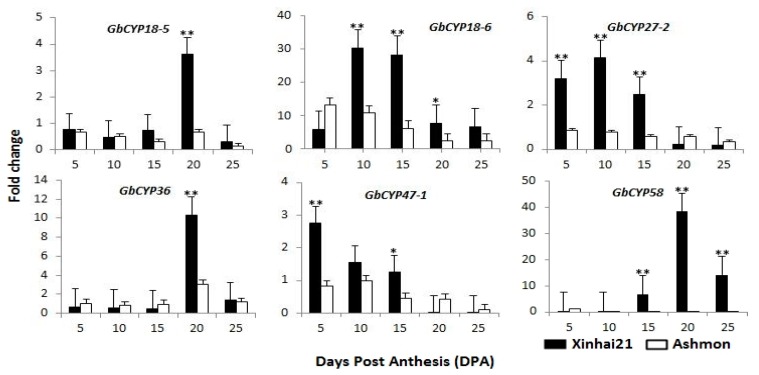
Expression profiles of six *GbCYPs* between Xinhai21 and Ashmon based on qRT-PCR. *GbUBQ7* was used as an internal reference to normalise the expression data. Error bars were calculated from the difference in the expression patterns of three biological replicates. The asterisks and double asterisks indicate correlations at the 0.05 and 0.01 significance levels, respectively (t-test, * *p* < 0.05, ** *p* < 0.01).

**Table 1 ijms-20-00349-t001:** Average cotton fibre quality parameters from 2014 to 2016.

Cotton	Fibre Length	Fibre Uniformity	Break ng Tenacity	Fibre Elongation	Short Fibre Rate	Maturity	Micronaire Value	Spinning Index
	mm	%	CN/tex	%	%			
Xinhai21	36.15 ± 0.49	87.83 ± 0.24	34.80 ± 0.58	6.10 ± 0.56	5.73 ± 1.44	0.87 ± 0.01	4.10 ± 0.12	165.00 ± 2.83
Ashmon	25.16 ± 0.56	81.20 ± 0.96	28.63 ± 2.00	9.06 ± 1.29	11.26 ± 3.60	0.85 ± 0.01	5.30 ± 0.23	104.33 ± 4.80

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
