# Peer review of "Genome-Wide Identification of Cyclophilin Gene Family in Cotton and Expression Analysis of the Fibre Development in Gossypium barbadense"

_ijms, 2019, doi:10.3390/ijms20020349_

Round 1

Reviewer 1 Report

In this manuscript, the author did Genome-Wide Identification of Cyclophilin Gene Family in Cotton and Expression Analysis of the Fiber Development in Gossypium barbadense. The manuscript is well written and will attract large readership. However, for the betterment of this manuscript, I have some comments to make.

Major:

1.      It will be nice it author include at least one Cyclophilin Gene characterization for fiber development.

2.      Add some recent genome-wide studies in this article such as:

a.      Genome-Wide Identification and Analysis of Genes, Conserved between japonica and indica Rice Cultivars, that Respond to Low-Temperature Stress at the Vegetative Growth Stage.

b.      Genome-Wide Analysis of the PYL Gene Family and Identification of PYL Genes That Respond to Abiotic Stress in Brassica napus.

c.      Genome-Wide Identification and Classification of the AP2/EREBP Gene Family in the Cucurbitaceae Species.

d.      Genome-wide analysis of gene expression profiling revealed that COP9 signalosome is essential for correct expression of Fe homeostasis genes in Arabidopsis.

e.      Genome-Wide Identification of the Dehydrin Genes in the Cucurbitaceae Species.

Minor:

1.      Change fibre development to fiber development in all manuscript.

2.      14 time “in addition” used in this manuscript. Please uses some other words.

3.      From p39 tp p44 6 times “have been” used. Please adjust this.

Author Response

Dear reviewer,

Thank you very much for carefully reviewing my article. Based on your comments, I have provided the following responses.

Major:

Point 1: It will be nice it author include at least one Cyclophilin Gene characterization for fiber development.

Response 1: Thank you very much for your suggestion. Based on our RNA-Seq and protein mass spectrometry data for Xinhai21, we have identified GbCYP18-6 and GbCYP27-2 as candidate genes for further investigation of CYP gene functions in cotton fibre development. Some results have been obtained, but they are not perfect, we are doing our utmost to confirm it.

Transgenic analyses show that trichome densities are increased. The roots are longer, and a larger number of root hairs than wild type are formed in Arabidopsis thaliana transfected with GbCYP18-6 and GbCYP27-2.

Point 2: Add some recent genome-wide studies in this article such as:

a. Genome-Wide Identification and Analysis of Genes, Conserved between japonica and indica Rice Cultivars, that Respond to Low-Temperature Stress at the Vegetative Growth Stage.

b. Genome-Wide Analysis of the PYL Gene Family and Identification of PYL Genes That Respond to Abiotic Stress in Brassica napus.

c. Genome-Wide Identification and Classification of the AP2/EREBP Gene Family in the Cucurbitaceae Species.

d.  Genome-wide analysis of gene expression profiling revealed that COP9 signalosome is essential for correct expression of Fe homeostasis genes in Arabidopsis.

e. Genome-Wide Identification of the Dehydrin Genes in the Cucurbitaceae Species.

Response 2: Five articles have been added in lines 69-80 follows:

Recently, as more and more plant genome data were published, genome-wide identification and analysis become an effective method for rapidly predicting gene functions from a large family of genes. There are an increasing number of reports about plant gene families. For example, genome-wide identification and expression analysis of the chitinase gene family in Brassica rapa revealed that it plays a crucial role in the pathogen resistance of the host plants[26]; genome-wide identification and analysis of japonica and indica rice cultivars identified 502 candidate low-temperature-responsive genes and proposed a model that includes a pathway for cold stress-responsive signalling[27]; genome-wide analysis of gene expression profiling indicated that the COP9 signalosome is essential for correct expression of Fe homeostasis genes in Arabidopsis[28]; genome-wide analysis of small RNAs revealed 257 novel microRNAs related to cotton fibre elongation[29], and genome-wide analysis of the WD40 protein family revealed that WD40 is involved in cotton fibre development [30,39].

Minor:

Point 1: Change fibre development to fiber development in all manuscript.

Response 1: I have Changed.

Point 2: 14 time “in addition” used in this manuscript. Please uses some other words.

Response 2: I have Changed “in addition” to “along with” in lines 24 and 58.

Point 3: From p39 tp p44 6 times “have been” used. Please adjust this.

Response 3: I have adjusted “have been” in lines 41 and 45.

Reviewer 2 Report

This is a well performed and clearly written bioinformatic account of the cyclophilins of cotton. It deserves to be published to add to information about this important crop.

Some specific comments:

The relationship between Xinhai21 and Ashmon cultivars should be stated clearly in discussion about results for the sea-island cotton fibres at line 217 - 227. This would make clearer what similarities and differences could be expected from their descent. 

CYP is a confusing abbreviation since it has been used for cytochrome P450s for decades and unfortunate that it was also selected for cyclophilins in 2004. If there is any opportunity to change to something else, it should be taken. 

The authors are confident that cyclophilins are implicated in cotton fibre development. The overview of the (limited) role for these proteins in plants is as part of signalling pathways in a large variety of situations (as is the case in other eukaryotes). Knowledge of signalling pathways in plants is highly biased towards a limited number of processes.  Should they be considered as typical signal pathway components rather than involved in specific abiotic and abiotic responses? 

The expression patterns in specific tissues is stronger evidence for roles of particular cyclophilins in these specific locations and developmental processes. 

Author Response

Dear reviewer,

Thank you very much for carefully reviewing my article. Based on your comments, I have provided the following responses.

Point 1: The relationship between Xinhai21 and Ashmon cultivars should be stated clearly in discussion about results for the sea-island cotton fibres at line217-227. This would make clearer what similarities and differences could be expected from their descent.

Response 1: I had added the relationship between Xinhai21 and Ashmon cultivars in lines 230-233 follows:

Additionally, Xinjiang is the only long-staple cotton-producing area in China. Xinhai21 is a hybrid of the Giza variety from Egypt and has been the major cultivar in Xinjiang over the years, and Ashmon is a resource material that was introduced from Syria. Our research team investigated the quality of these two sea-island cotton fibres over the 3 years under field conditions and found that Xinhai21 had a greater fibre length than Ashmon (Table 1).

Table 1  Average cotton fibre quality parameters from 2014 to 2016

cotton

Fibre   length

Fibre   uniformity

Break   ng tenacity

Fibre   elongation

Short   fibre rate

Maturity

Micronaire vaule

Spinning index

·mm

·%

·CN/tex

·%

·%

Xinhai21

36.15±0.49

87.83±0.24

34.80±0.58

6.10±0.56

5.73±1.44

0.87±0.01

4.10±0.12

165.00±2.83

Ashmon

25.16±0.56

81.20±0.96

28.63±2.00

9.06±1.29

11.26±3.60

0.85±0.01

5.30±0.23

104.33±4.80

Point 2: CYP is a confusing abbreviation since it has been used for cytochrome P450s for decades and unfortunate that it was also selected for cyclophilins in 2004. If there is any opportunity to change to something else, it should be taken.

Response 2: This is also a confusing question for me. I do not think I can change its name for the following two reasons.

First, Handschumacher et al. first identified cyclophilin A from bovine thymocytes in 1984 and named it CyPA. Subsequently, researchers have used such terms for all animals, particularly humans, such as, human CYP A (hCYP-A), hCYP-B, hCYP-C, hCYP-D, hCYP-E, hCYP-40 and hCYP-NK.

Second, in plants, CYP is also used to name cyclophilin family genes, such as Arabidopsis thaliana (AtCYP), rice (OsCYP), soybean (GmCYP), Brassica napus (BnCYP), maize (ZmCYP), and tomato CYP[1-6].

1.       Romano, P.G.; Horton, P.; Gray, J.E. The Arabidopsis cyclophilin gene family. Plant Physiol. 2004, 134, 1268-1282.

2.       Mainali, H.R.; Chapman, P.; Dhaubhadel, S. Genome-wide analysis of Cyclophilin gene family in soybean (Glycine max). BMC plant biology. 2014, 14, 282.

3.       Hanhart, P.; Thieß, M.; Amari, K.; Bajdzienko, K.; Giavalisco, P.; Heinlein, M.; Kehr, J. Bioinformatic and expression analysis of the Brassica napus L. cyclophilins. Scientific Reports. 2017, 7, 1514.

4.       Ahn, J.C.; Kim, D.W.; You, Y.N.; Seok, M.S.; Park, J.M.; Hwang, H.; Kim, B.G.; Luan, S.; Park H.S.; Cho, H.S. Classification of rice (Oryza sativa L. Japonica nipponbare) immunophilins (FKBPs, CYPs) and expression patterns under water stress. BMC Plant Biology. 2010, 10, 253.

5.       Wang, Q.; Wang, Y.; Chai, W.; Song, N.; Wang, J.; Cao, L.; Jiang, H.; Li, X. Systematic analysis of the maize cyclophilin gene family reveals ZmCYP15 involved in abiotic stress response. Plant Cell, Tissue and Organ Culture (PCTOC). 2017, 128, 543-561.

6.       Gasser, C.S.; Gunning, D.A.; Budelier, K.A.; Brown, S.M. Structure and expression of cytosolic cyclophilin/peptidyl-prolyl cis-trans isomerase of higher: plants and production of active tomato cyclophilin in Escherichia coli. Proceedings of the National Academy of Sciences. 1990, 87, 9519-9523.

Point 3: The authors are confident that cyclophilins are implicated in cotton fibre development. The overview of the (limited) role for these proteins in plants is as part of signalling pathways in a large variety of situations (as is the case in other eukaryotes). Knowledge of signalling pathways in plants is highly biased towards a limited number of processes. Should they be considered as typical signal pathway components rather than involved in specific abiotic and abiotic responses?

Response 3: Upon consulting the relevant reference, I think it is true that cyclophilins have an essential function in signalling. For instance, Giavalisco et al. identified 140 soluble phloem proteins of B. napus by 1-DE and high-resolution 2-DE, which included 9 CYPs protein[7], and Hanhart et al. performed LC-MS/MS analysis of phloem protein extract of B. napus and found 12 BnCYPs in the phloem[8]. Current research has confirmed that the phloem constitutes a long-distance transport system for trafficking plant hormones and signals and for the distribution of nutrients throughout the plant.

In addition, while investigating the dgt (auxin-resistant diageotropica) mutant of tomato and using Fourier transform infrared spectroscopy, it was discovered that cyclophilin (LeCYP) is a component of auxin signalling [9]. Yeast two-hybrid and pull-down assays demonstrated that AtCYP20-2 interacted with BZR1 and changed its conformation to regulate the activity of Brassinolide (BR)[10,11].

However, the evidence is limited at present, and the real mechanism of CYPs remains to be studied.

7.       Giavalisco, P.; Kapitza, K.; Kolasa, A.; Buhtz, A.; Kehr, J. Towards the proteome of Brassica napus phloem sap. Proteomics. 2006, 6, 896-909.

8.       Oh, K.C.; Ivanchenko, M.G.; White, T.J.; Lomax, T.L. The diageotropica gene of tomato encodes a cyclophilin: a novel player in auxin signaling. Planta. 2006, 224, 133-144.

9.       Zhang, Y.; Li, B.; Xu, Y.; Li, H.; Li, S.; Zhang, D.; Mao, Z.W.; Guo S.Y.; Yang C.H.; Weng Y.X; et al. The cyclophilin CYP20-2 modulates the conformation of BRASSINAZOLE-RESISTANT1, which binds the promoter of FLOWERING LOCUS D to regulate flowering in Arabidopsis. The Plant Cell. 2013, tpc-113.

10.    Sun, Y.; Veerabomma, S.; Abdel-Mageed, H.A.; Fokar, M.; Asami, T.; Yoshida, S.; Allen, R.D. Brassinosteroid regulates fiber development on cultured cotton ovules. Plant and Cell Physiology. 2005, 46, 1384-1391.

11.    Zhang, M.; Zheng, X.; Song, S.; Zeng, Q.; Hou, L.; Li, D.; Zhao, J.; Wei, Y.; Li, X.B.; Luo, M. Spatiotemporal manipulation of auxin biosynthesis in cotton ovule epidermal cells enhances fiber yield and quality. Nature biotechnology. 2011, 29, 453.

Point 4: The expression patterns in specific tissues is stronger evidence for roles of particular cyclophilins in these specific locations and developmental processes.

Response 4: Thank you for your affirmation. These results provide important clues for my subsequent experiments.

Based on our RNA-Seq and protein mass spectrometry data for Xinhai21, we have identified GbCYP18-6 and GbCYP27-2 as candidate genes for further investigation of CYP gene functions in cotton fibre development. Some results have been obtained, but they are not perfect, we are doing our utmost to confirm it.

Transgenic analyses show that trichome densities are increased. The roots are longer, and a larger number of root hairs than wild type are formed in Arabidopsis thaliana transfected with GbCYP18-6 and GbCYP27-2.

Round 2

Reviewer 1 Report

I am happy with the reply of author. This manuscript can be accepted in its current format.